# Identification of Skeletal Remains Using Genetic Profiling: A Case Linking Italy and Poland

**DOI:** 10.3390/genes14010134

**Published:** 2023-01-03

**Authors:** Francesca Tarantino, Luigi Buongiorno, Benedetta Pia De Luca, Alessandra Stellacci, Michele Di Landro, Gabriele Vito Sebastiani, Gerardo Cazzato, Stefania Lonero Baldassarra, Emilio Nuzzolese, Maricla Marrone

**Affiliations:** 1Section of Legal Medicine, Department of Interdisciplinary Medicine, University of Bari “Aldo Moro”, 70124 Bari, Italy; 2Section of Molecular Pathology, Department of Emergency and Organ Transplantation (DETO), University of Bari “Aldo Moro”, 70124 Bari, Italy; 3Section of Legal Medicine, Department of Public Health Sciences and Pediatrics, University of Turin, 10121 Turin, Italy

**Keywords:** skeletal remains identification, forensic genetics, Prüm convention, forensic pathology

## Abstract

Forensic genetics is a rapidly evolving science thanks to the growing variety of genetic markers, the establishment of faster, less error-prone sequencing technologies, and the engineering of bioinformatics models, methods, and structures. In the early 2000s, the need emerged to create an international genetic database for forensic purposes. This paper describes a judicial investigation of skeletal remains to identify the subject using various methods. The anthropological examination of the remains allowed identification of the Caucasoid (European) ethnic group, a height of 156 ± 4 cm, and an age between 47 and 50 years. The genetic profiles obtained from typing several microsatellites made it possible to evaluate the compatibility between the skeletal remains and the suspected decedent. To identify the remains, the two extrapolated genetic profiles were compared. The case described highlights the central role of forensic genetics in identifying skeleton remains by means of comparison.

## 1. Introduction

Genetic investigations were first applied in the forensic anthropological field in the 1980s in the study of anomalous and uncontrollable migratory flows, and in particular of criminal acts, catastrophes of all kinds, and war events, and the need for forensic investigation of skeletons or unrecognizable bodies [1,2]. Genetic studies of cadaveric remains have made it possible to acquire information, in forensic contexts, regarding remains found and, in general, about migrations, human evolution, and population genetics [3].

In the forensic field, the most commonly used genetic markers are microsatellites or STR (short tandem repeats). A comparison between two genetic profiles obtained by typing several microsatellites makes it possible to evaluate the compatibility between the two subjects [4,5]. The principle behind these studies lies in comparing some markers on the hypervariable sequences corresponding to 0.3% of the entire DNA molecule.

Thanks to these comparisons, forensic genetics is now the cornerstone of some judicial investigations [1]. Sequencing techniques have allowed the identification of an ever-increasing number of genetic markers that are useful for forensic purposes [5]. Although sequencing of the human genome took more than twenty years, and more than fifty years of research with modern technologies have passed, it is now possible to obtain the complete sequence of a human genome in a few days [6]. Furthermore, the analysis of genetic material has been extended to other animal species, plants, and microorganisms [7].

Microbiome and metagenomic analyses are inspired by the fact that human beings host diverse microbial communities at the level of all body surfaces, continuously interacting with and altering the surrounding environments. The latest studies aim to study the information relating to these interactions, especially for thanato-chronological purposes [8].

In line with the advances in these technologies, in the early 2000s, the need emerged to create an international genetic database for forensic purposes [9]. This need was then formalized through the Prüm treaty signed on 27 May 2005 by seven member states of the European Union (Germany, Spain, France, Austria, Belgium, the Netherlands, and Luxembourg) to strengthen police cooperation in the fight against terrorism, cross-border crime, and illegal immigration. Specifically, the Prüm Agreement, in addition to providing for the exchange of data relating to DNA and fingerprints, promotes exchanging information on the subject, referring to persons who have disappeared or have suffered violence [10].

It follows that with the advances made in meta-genomic technology and computational prediction, as well as the creation of international genetic databases, possible issues related to individual privacy have emerged. However, this subject will not be dealt with in depth in this manuscript.

This paper describes a judicial investigation of skeletal remains to identify the subject using various methods. Personal identification was made possible by comparing the genetic profile from the skeletal remains with a genetic profile from a sample obtained from a previous sexual assault victim.

The manuscript aims to highlight the complexity of forensic investigations in human skeletal remains, with particular attention to the cornerstone role of forensic genetics in attributing identity to skeletal remains, per se limited by the absence of an adequate comparator database.

## 2. Material and Methods

The first investigation carried out was that of the inspection. On that occasion, metric and photographic surveys were carried out on the remains, which were then transported to the Forensic Anthropology laboratory of the University of Bari “Aldo Moro.” At the same time, investigations by the Judicial Police were launched into the environment of neglected and homeless people, narrowing the field around an unknown subject, a prostitute and the victim of multiple episodes of abuse, including sexual violence, that had led to the collection of biological samples, appropriately stored in the laboratories of the Scientific Section of the Police.

For recognition, several investigations were carried out.

### 2.1. Forensic Anthropological Investigations

A 1-milligram bone sample (femur) was taken for the dating study of bone remains using an analysis of the carbonate phosphate (C/P) index by an external laboratory. 

The anthropological measurements were performed according to international standards of forensic anthropology [11,12] and were aimed at determining ethnicity and sex, estimating height, and determining age, as described below in each section.

#### 2.1.1. Determination of Ethnicity

For the determination of the ethnic group, the following parameters of the skull were examined:sagittal contour;facial morphology;frontal and parietal bones;orbits;interorbital distance;nasion;nasal opening;nasal spine;zygomatic arches;palate;mastoid;nuchal crest.

#### 2.1.2. Sex Determination

The following skeletal segments were examined for sex determination:morphological and craniometrical parameters of the skull;morphological parameters of the pelvis.

#### 2.1.3. Stature Determination

The remains were analyzed following the method proposed by Ozaslan, using the right humerus, radio, femur, and tibia [11,12,13]

#### 2.1.4. Age Determination

The Suchey-Brooks method [14], the method by Meindl, Lovejoy, and Mensforth [15], and the Derobert-Fully method [16] were used for the evaluation of the skeletal remains.

### 2.2. Investigations of Forensic Dentistry

The dental arches were reconstructed with the dental elements found, and the teeth, the upper jaw, and the mandible were then inspected with exposure to a source of UVA light (X5, Inova). Periapical digital photographic and radiographic surveys of the dental arches and sensors for radiovideographies were made [17,18,19,20].

### 2.3. Investigations of Forensic Genetics

To isolate the DNA, allowing personal identification of the subject whose skeletal remains had been found, two teeth (34, 37) and a 1 cm piece of the femoral shaft of the skeleton were removed.

The vaginal swab taken during a previous judicial investigation into a case of sexual violence, and duly filed in the refrigerators of the Scientific Police, was later used for comparative purposes to verify whether the remains found could belong to a previous victim of sexual abuse.

Tooth 37 and the vaginal swab were subjected to DNA extraction at the Forensic Genetics laboratory of the University of Bari, using the commercial kit NucleoSpin^®^DNA Tissue by the company MACHEREY-NAGEL, Allentown, PA, USA, following the specific protocols indicated by the manufacturer [21,22].

In particular, in the preliminary stages of the investigation, the DNA was extracted and examined starting from tooth 37; after the investigations conducted by the Judicial Police, it was possible to access the vaginal tampon. DNA extraction from the vaginal swab was subsequently performed by a female operator with all the PPE provided and after the decontamination of the worktops, pipette tips, and pipes, with exposure to UV rays, sodium hypochlorite, and ethanol.

### 2.4. Investigations of Forensic Radiology

Skull CT was performed.

## 3. Results

### 3.1. Inspection Investigations

The skeletal remains were found on the fourth floor of an abandoned farmhouse in the industrial area of a city in Southern Italy, partially covered by a wooden box and some wooden planks, arranged as a funeral bed (Figure 1).

The distribution of the bone remains, with some residues of mummified soft tissues, suggested that the corpse had been laid in supine decubitus, with the upper limbs flexed and adducted to the trunk and the lower limbs in normal position.

The skeleton was wrapped with two pieces of duct tape (Figure 2).

The right shoulder bones (clavicle, humerus, and scapula) were wrapped in a cotton fabric.

There were signs of micro- and macrofauna activity on the skeletal remains and in the surrounding environment.

### 3.2. Examination of the Skeletal Remains

The carbonate phosphate index (C/P) identified the skeletal remains as recent clinical/forensic bone samples (C/P = 0.66) and not as archaeological remains. The remains were carefully cleaned and placed in an anatomical position.

The skeleton appeared complete and intact, light brown in color, with bones.

−The Skull: Intact and complete with a fractured right portion of the nasal bone; dental elements and partial dentures were found.−Hyoid bone: The body and the large horns were intact but disjointed.−Spine: All vertebrae were present and intact.−Clavicle and scapula: The joint complexes were intact and inter-articulated.−The ribs and the sternum were intact and inter-articulated.−Pelvis: The pelvis was intact but disjointed from the sacrum.−Upper limbs: Humeri intact and preserved, the left disjointed. Ulna and radius were present and intact. The bones of the complete right hand, the pisiform bone, the bones of the distal row of the carpus, and the distal phalanges of the IV-V radius of the left hand were missing.−Lower limbs: Femur, patella, and tibia present bilaterally. The soles of both feet were complete.

### 3.3. Determination of Ethnicity

To determine ethnicity, the morpho-dimensional characteristics of the skull were examined, and the distances summarized in the following table calculated (Table 1):

### 3.4. Determination of Sex

To determine the sex, the following skeletal remains were examined:−Morphological parameters of the skull, which appeared long and not very robust, with thin and sharp supraorbital, small, rounded, and medially inclined mastoid processes, pronounced frontal and parietal bones, spherical orbit with sharp edges, forehead, regular and U-shaped palate, nuchal crest and insertion crests of the nuchal muscles, glabella, cheekbones, and minimal development of mastoids.−Pelvis, macroscopically, appeared low and wide with a broad, heart-shaped pelvic cavity. The subpubic angle was shaped like an inverted U. The preauricular sulcus was present, the auricular surface raised and stepped, the ventral arch was present, the obturator foramen was triangular, the ischio-pubic branch moderately thin and narrow, the sacrum flattened in an anteroposterior direction.

These elements depicted the skull and pelvis of a female subject.

### 3.5. Determination of the Stature

Measurement of the right femur yielded a maximum length of 40 cm, equal to a stature of 156 ± 4 cm.

### 3.6. Age Determination

#### 3.6.1. Suchey-Brooks Method

The Suchey-Brooks method was used to determine the age of the subject using morphological analysis of the modifications of the symphysis facet of the pubis to define the age of a white female cadaver. The articular surface was free of crestlines, flat, with evidence of a slight depression of the symphysis facet and lipping (presence of minute lingulae) on the dorsal margin; bony outgrowths of the ventral margin were found, with signs of bony destruction on its upper margin.

The application of this method showed a presumed age of 48.1 ± 14.6 years.

#### 3.6.2. Meindi, Lovejoy, and Menshorth Method

This method, which studies age-related changes in the auricular surface of the ileum, allowed us to highlight a skeletal age between 47 and 54.5 years.

#### 3.6.3. DeRobert-Fully Method

This method, which evaluates the degree of obliteration of the exocranial sutures, allowed us to detect complete obliteration of the sagittal suture, an almost complete obliteration of the coronal suture near the first and third segments, and partial obliteration of the lambdoid suture in correspondence with the first and second section. These data indicated an estimated age of 45–50 years.

The confluence of the various methods applied made it possible to place the average skeletal age between 47.5 and 50.75 years.

### 3.7. Investigations of Forensic Radiology

Examination of the skull made it possible to detect a displaced bone fracture of the right nasal bone (antemortem) with a small, displaced bone fragment.

There was no evidence of post-traumatic alterations of the remaining bone structures of the skull.

### 3.8. Investigations of Forensic Dentistry

The characteristics of the skull and jaw allowed us to recognize a typical conformation of the female sex and Caucasoid ethnicity.

The study of the relationships between the extension of the dental pulp and the volume of the entire tooth, obtained with radiographic images of the upper and lower canines according to Cameriere’s formula, allowed us to confirm an estimated age equal to 50.4 (±5) years [23].

The general state of the mouth, wear of dental prostheses, and dental neglect made it possible to conclude that the subject had not undergone dental checks for at least five years.

### 3.9. Investigations of Forensic Genetics

Research on DNA polymorphisms established that the DNA extracted from tooth 37 and the vaginal swab possessed the genetic characteristics summarized in the following table (Table 2). Only peaks with an RFU > 50 were considered, and the dropout in D7S820 and TPOX markers is a hypothesis supported by the fact that TPOX showed a peak at allele 9 with an RFU < 50 in both replicates of DNA PCR extracted from tooth 37.

## 4. Discussion

The judicial investigation described in the study was aimed at identifying the individual whose skeletal remains had been found.

Personal identification of the remains of an unknown subject is obtained through visual identification by showing the body or photographic reproduction of the face to people they knew in life [24]. When, on the other hand, the cadaveric transformation processes (putrefaction, maceration, mummification, adipocerization, corification) or skeletonization of the remains (action of the micro- and macrofauna, carbonization, decay, etc.) do not allow this, the ascertainment of identity requires specific investigations with a multidisciplinary approach to limit the risk of interpretative bias [25].

In an abandoned cadaver, tissue transformation is completed on average within 3–5 years. Gradually, after the destruction of the soft parts, the fibrous tendon and cartilage tissues disappear, and only the bones remain. Over time, the bones are freed from even the most minor residues of soft parts that are still adhered, dried, or encrusted but retain their organic component for a long time. Later, after many years, only mineral scaffolding made of phosphates and calcium carbonates remains. Still, although decalcification takes place with the passing of time, bones remain light, porous, brittle, and susceptible to pulverization at the slightest contact [26]. However, it should be remembered that the skeletonization process is strongly influenced by the environmental conditions where the corpse resides, as well as by the activity of cadaveric micro- (insects) and macro-fauna, which can accelerate the removal of all soft tissues as well as cause mutilation to or dismemberment of the body [26]. Adopting rigorous objectivity criteria capable of standardizing the logical and final decision-making processes is fundamental for identification investigations. The standardized operating protocols, therefore, provide the preliminary definition of generic identification characteristics (ethnicity, sex, age, height, etc.) capable of ascertaining the anthropometric connotations of the individual to be identified. Subsequently, we search for elements with high individual specificity and that are highly characterizing, which can be the object of helpful comparisons with similar findings pertinent to the presumed victim when alive. Whenever possible, personal identification is based on comparing the genetic profiles obtained from the corpse with those extracted from presumed family members and objects that belonged to the victim [1]. The inspection did not offer any information regarding the corpse’s identity in the case in question. The carbonate phosphate index (C/P) confirmed the skeletal remains to be recent clinical/forensic bone samples. The anthropological examination of the remains allowed identification of the Caucasoid ethnic group, female sex, a height of 156 ± 4 cm, and an age between 47 and 50 years. The information acquired through anthropometric studies, as well as the results of the forensic dental studies, compared with the anthropometric characteristics obtained by the police of female subjects who had disappeared in the last period in Puglia, did not allow the identification of the cadaver, so DNA extraction and typing were started from one molar (37).

One of the main limitations of forensic samples, in addition to the fact that they contain only small amounts of biological material, is that very often the DNA is degraded, i.e., reduced into small fragments by chemical and physical agents that cause the rupture of the ligaments of the double helix [27,28,29,30].

In the present case, obtaining the generic DNA profile from the two molars was possible without using extraction from the femur. In particular, one of the limitations of our study was the limitation imposed by the magistrate in charge of the case, who expressly requested we protect the most significant number of samples for any subsequent forensic investigations.

A study of the genetic marker “amelogenin” identified the sex of the subject, specifically confirming what was inferred from the forensic anthropological study, namely that the individual was female.

In the case of our study, once the genetic profile had been obtained from the no. 37 molar, it was isolated and then compared with the data of a profile extracted from a vaginal sample of a woman who had suffered sexual violence a few years before the discovery of these skeletal remains. This comparison was made because, during the inspection, the police had noticed the recurrence of a name written with a spray can on the farmhouse wall where the remains were found.

However, after consulting the missing person reports, the police could not trace any individual with that name. The agents then extended the search to the regional register of reported episodes of violence, tracing back to a woman whose name in Polish was similar to the one written on the wall.

The woman listed in the regional register was Polish and had reported sexual violence; the emergency room health workers, on that occasion, had carried out a vaginal swab from which the genetic profile was extrapolated.

To identify the remains, the two extrapolated genetic profiles were compared and determined to be statistically compatible, allowing the identification of the subject with the genetic profile observed in tooth 37, excluding the markers that most likely showed the dropout phenomenon (D7S820 and TPOX).

Considering the possible compatibility of the two genetic profiles, various investigations were carried out by the national and international police forces. Thanks to these investigations, it was learned that the woman, aged 51 years old, from Poland, had gone to Italy in the year preceding the discovery of the skeletal remains and was involved in a prostitution ring. From all the anthropological elements that emerged, it was possible to attribute the cause of death to a traumatic injury resulting from aggression.

It is interesting to note that genetic profiling was carried out using a relatively recent low-cost technology.

Forensic genetics is a rapidly evolving science thanks to the growing variety of genetic markers, the establishment of faster, less error-prone sequencing technologies, and the engineering of bioinformatics models, methods, and structures [28].

In our case, the identification was possible thanks to “luck” in finding a clue that allowed the police to carry out in-depth research, helping us to match the genetic profile isolated from the skeletal remains with one previously collected from a victim of violence.

If a match had not been found, the next step would have been to consult the international databases, as required by the Prüm agreement [10].

In fact, for the genetic information relating to an individual to be entered into the international database, precise sampling and analysis methods of the biological find must be followed according to protocols validated at the international level and indicated by the European Network of Forensic Science Institutes (ENFSI) [29]. The need to adhere to protocols defined by multinational companies is a form of protection for individuals, given the delicacy and confidentiality of the field of application. Incorrect storage of the artifact, incorrect processing of the material available, or careless and forced conclusions about the results can alter the outcome of the proceedings and fail to safeguard the dignity and freedom of the individual involved. It is often impossible to carry out analytical counter-tests due to the often-limited quantity of forensic biological findings [31,32,33,34,35,36].

However, the addition of genetic information is allowed only in a few forensic laboratories that follow specific certification procedures controlled by external bodies. Alternatively, authorization from the Judicial Authority is required.

In addition, there are restrictions on consultation so as to safeguard the individual’s privacy by allowing access to data only by a limited number of laboratories.

## 5. Conclusions

The case described illustrates the central role of forensic genetics in identifying skeletal remains using comparison. However, surveys on an international scale still present limits to database consultation due to legal and political implications related to privacy and protected health information. The search for a missing person is challenging for law enforcement officers and requires collaboration between states. The authors propose the facilitation of access to international databases without neglecting ethical issues or the privacy of the individual. On the contrary, the individual must remain at the center of future philosophical, sociological, and legal work while introducing further provisions to make identification more accessible internationally.

## Figures and Tables

**Figure 1 genes-14-00134-f001:**
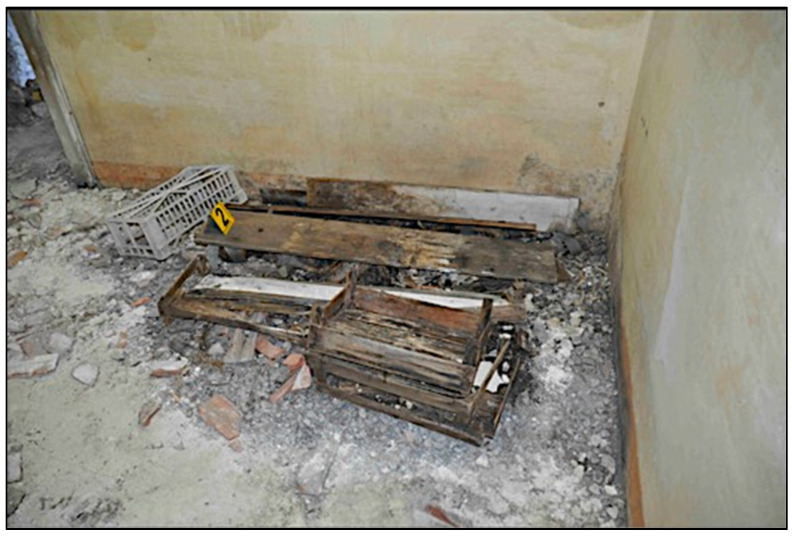
Place of discovery of the skeletal remains.

**Figure 2 genes-14-00134-f002:**
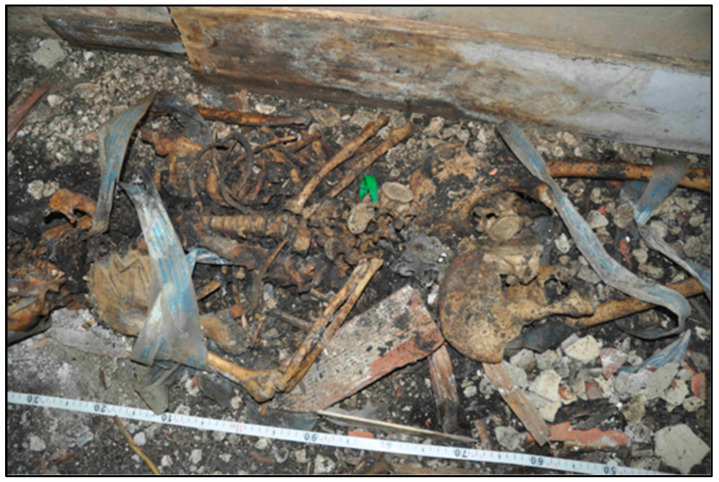
Arrangement of the remains at the time of discovery.

**Table 1 genes-14-00134-t001:** Anthropometric measurements of the skeletal remains.

Distance	Centimeters
Basion-Prosthion	8.9
Glabella-Opistocranion	16.8
Basion-Nasion	10.3
Prosthion-Nasion	6.9
Nasal height	4.8
Max width	14.5
Basion-Bregma	13.4
Bizigomatic	11
Nasal amplitude	2.2

**Table 2 genes-14-00134-t002:** Genetic markers were extracted from tooth No. 37 and a vaginal swab.

GENETIC MARKERS	VAGINAL SWAB	TH. 37
D8S1179	**13/14**	**13/14**
D21S11	**30/30.2**	**30/30.2**
D7S820	**8/9**	**9**
CSF1P0	**11/12**	**==**
D19S433	**13/14.2**	**13/14.2**
VWA	**16/16**	**16/16**
TPOX	**8/11**	**11**
D18S51	**13/15**	**==**
D3S1358	**15/16**	**15/16**
TH01	**8/9**	**8/9**
D13S317	**9/12**	**9/12**
D16S539	**12/13**	**12/13**
D2S1338	**16/24**	**==**
D5S818	**12/12**	**12/12**
FGA	**19/21**	**19/21**
Amelogenina	**X/(Y)**	**X/X**

== RESULT ABSENT OR NOT INTERPRETTABLE.

## Data Availability

Not applicable.

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
