# Peer review of "Identification of Skeletal Remains Using Genetic Profiling: A Case Linking Italy and Poland"

_genes, 2023, doi:10.3390/genes14010134_

Round 1
Reviewer 1 Report (New Reviewer)
Dear Authors,
Nice and well-done work. The presentation was very good and understandably. Just one note - you should write the reason why you compared to the vaginal tampon at the beginning (in the paragraph where you first mentioned the vaginal tampon)
Author Response
Dear Reviewer n'1,
thank you very much for your congratulations. We added a specific answer in the paper.
A warm greeting
Reviewer 2 Report (New Reviewer)
Overall lack of clarity, in particularly on defining objectives and lack of match between objectives and results; some approaches presented in the section Methods have no corresponding results, such as statistical calculations.
Inappropriate referencing/citation (e.g., line 403: as already widely discussed in Nature in 2013)
Either due to poor English language or scientific ignorance, many incorrect statements (e.g., line 38: STR (short tandem repeats) composed of temporary units of nucleotides repeated in tandem; line 405: The authors believe that it is not the limitation of access to the consultation that protects the individual but the implementation of privacy protections different from those currently existing without limiting access to this data to industry experts).
Only the genetic part is suitable for this journal; more appropriate to a forensic one.
Author Response
Dear Reviewer n'2, We are very sorry that you did not like our paper. However, we corrected according to all his indications. Thank youReviewer 3 Report (New Reviewer)
Dear Authors, I enjoyed reading your manuscript and made some editorial suggestions for your consideration that should help the flow and content of the paper. Other than the minor editorial comments/suggestions that I made to your manuscript, please see if you can combine some of the "single sentence paragraphs" to make it easier to follow and read. I found all of your methods appropriate, but would suggest providing a little more detail about the methods used and conclusions for the decedent's "biological profile" (age, ancestry/ethnicity, sex and stature). You might also expand a little on your reliance of the molar(s) for estimating age at death and how that compares with the skeletal methods that you used and if they are consistent or not. Also, please see my comments on your descriptions and choice of some words of the skeletal features that you used. I enjoyed reading your manuscript, found it interesting.
If possible, I would suggest including a photograph of the entire skeleton laid out in anatomical position/order for examination, as many readers will want to see the features of the skull and pelvis (features used for estimating the biological profile) for themselves. A skeletal photograph would also give the reader a better understanding of the condition of the remains. Lastly, some readers might like to know why you chose the craniometrics that you did and not the whole battery of measurements that, when used together, typically yield a stronger result for sex and ancestry. Thank you again for all your hard work in this case.

Author Response
Dear Reviewer n'3, first of all thank you very much for your patience, your thoughtfulness and your valuable advice aimed at improving the quality of our manuscript. We followed all his precious indications step by step, which allowed us to improve the quality of our paper. We have not only managed to recover a photo of the skeleton in an anatomical position, but we have added all the details on the methods used for the evaluation of the anthropometric and age/sex/ethnicity measurements. Let's hope it lives up to it.Round 2
Reviewer 2 Report (New Reviewer)
The new version has not answered my previous questions (except two, solved by removals).
Namely,
1. lack of clarity, in particularly on defining objectives and lack of match between objectives and results remain
2. some approaches presented in the section Methods have no corresponding results, such as statistical calculations and serious errors persist, such as: line 169 « calculation of the Random Match Probability (RMP) corresponds to the estimate of the frequency with which a profile occurs in the population»
Some References are no longer cited in the text (e.g. 33).
Text revision was careless, in some cases worsening the previous version; e.g.:
line 448 The case described has highlighted the central role forensic genetics assumes in identifying skeletal remains by comparison.
Either due to poor English language or scientific ignorance, many incorrect or incomprehensible statements persist, as in
line 397 " To identify the remains, the two extrapolated genetic profiles were compared, which were statistically compatible, allowing the identification of the subject. The genetic profile observed in tooth 37, excluding the markers that most likely showed the dropout phenomenon (D7S820 and TPOX), gave an RMP (Random match probability) equal to 2.9 and -12. Once the RMP result was obtained, the LR (Likelihood ratio) was also calculated, getting an effect similar to 353,814.78, there-fore higher than the standard threshold of 10,000..
Author Response
Dear Reviewer n'2,
first of all thank you very much, again, for the patience and for your tips useful to improve the quality of our manuscript.
Actually, we have completely modified the english language by a native-english speaker and review the entire manuscript according to the suggestions.
A warm greeting
This manuscript is a resubmission of an earlier submission. The following is a list of the peer review reports and author responses from that submission.
Round 1
Reviewer 1 Report
Overall this is a practical case presented in an easy to read and well organized manuscript. A more detailed description and analysis of the STR profiles of the case samples would be better.
1. Was the genotype of Amelogenina from vaginal swabs is x and y? Is the sample contaminated?
2. During the whole experiment, are there any special steps to prevent contamination?
3. How to determine whether there is allele dropout rather than homozygosity at these two loci (D7S820 and TPOX)?
Author Response
Dear Editor,
thank you for allowing us to submit a revised draft of our manuscript. We appreciate the reviewers’ suggestions very much. We have been able to make changes as suggested by the reviewers. We have highlighted the changes within the manuscript.
Below, you can find a point-by-point response to the reviewers’ comments.
Reviewer #1:
Reviewer’s comment: “Was the genotype of Amelogenina from vaginal swabs is x and y? Is the sample contaminated?”
Authors ‘comment: We are grateful to the reviewer for her/his positive comments and the chance to explain better what was highlighted. The genotype of Amelogenin extracted from the vaginal swab gave two peaks: x with RFU=4200 and y with RFU=274.
Reviewer’s comment: “During the whole experiment, are there any special steps to prevent contamination?”
Authors ‘comment: Thank the reviewer for the valuable advice on the reviewer's contamination during laboratory experiments; contamination is likely attributable to the alleged sexual violence suffered by the victim. Contamination during laboratory investigations is excluded as these were carried out according to the Italian Society of Human Genetics (SIGU) guidelines to prevent contamination of the sample. DNA extraction from the vaginal swab was performed individually and subsequently by a female operator with all the PPE provided and after decontamination of worktops, pipette tips, and pipes, respectively, with exposure to UV rays, sodium hypochlorite, and ethanol. We have specified this concept in the paper as follows: “All procedures have been performed in compliance with SIGU guidelines to prevent possible contamination of samples.”
Reviewer’s comment: “How to determine whether there is allele dropout rather than homozygosity at these two loci (D7S820 and TPOX)?”
Authors ‘comment: We are grateful to the reviewer for displaying a confusing element for readers that we have modified as follows: “Only peaks with RFU> 50 were considered, and the drop-out in D7S820 and TPOX markers is a hypothesis supported by the fact that TPOX showed a peak at allele 9 with RFU < 50 in both replicates of DNA PCR extracted from tooth 37. Since no samples of non-degraded tissue were available, it was not possible to confirm abandonment or homozygous in other samples.”
Reviewer 2 Report
- In my oppinion you must revise well all the material and methods. Here by some notes:
- In line 95, when you speak about Determination of ethnicity, you don not cite the authors of the methods used. In the same way, on line 110, when you speak about the sex anthroplogogical diagnosis, you neither cite any author. I think they must be named.
- In the material and methos of the genetic analysis I have many questions:
o You Indicate that: “to isolate the DNA than would allow us to personally identify the subject to whom the skeletal remains had belonged in life, two teeth (34, 37) and a 1cm piece of the femoral shaft of the skeleton were removed” (line 130). Nevertheless, on line 136, you indicate than only “tooth 37 and the vaginal swab were subjected to DNA extraction…”. The other skeletal pieces (tooth 34 and piece of femur) were not analyzed? Nevertheless, on the Discussion you indicate than the both dental pieces were analyzed. It is so confusing. I think it must be clarified.
o About the skeletal remains genetic analysis: On material and methods you indicate that the methodology used to extract DNA from the sample 37 and the vaginal swab was the same. It is very surprising to me, so to the DNA extraction from teeth or bones, usually is necessary to apply specific conditions and also specific cares to minimize contamination. You do not specify what kind of authenticity criteria have used to ensure the absence of exogenous contaminant DNA in the case of the critical skeletal samples. This kind of samples usually are analyzed in specific isolated laboratories, where are no analyzed other kind of well-preserved samples. The sample 37 and the vaginal swab were analyzed simultaneously in the same laboratory? According to the material and methods description it is not clear, and it could be interpreted as a possible source of contamination for the sample 37 (furthermore if the results were no replicated with the sample 34 or the femur). In my opinion, there are too many things to clarify in material and methods of the genetic analysis and the authenticity criteria considered to skeletal remains analysis.
o If there were analyzed more skeletal samples, How many extractions were peformed? And, how many PCRs from each extract?
o About the STRs employed you indicate that are “the amin Caucasian variant” (line 151), not only, are the considered on CODIS.
o (Line 167) about the first skeletal remains finding, I would like it there were performed some dating method to ensure that the remains are contemporary.
o About the results obtained, on Table 2:
§ I think that it would be explained how were stablished the consensus DNA profile in the case of the skeleton. If there were obtained by different PCRs from different samples (34, 37 and femur), How the consensus was stablished?
§ What was the criteria to define homozygotes? You consider that there was an allelic dropout phenomenon on: D7S820 and TPOX. Why the cases of: D5S818, and VWA are not also allelic dropout phenome? What was the criteria to stablish these homozygotes?
Author Response
Reviewer #2:
Reviewer’s comment: “In line 95, when you speak about Determination of ethnicity, you don not cite the authors of the methods used. In the same way, on line 110, when you speak about the sex anthroplogogical diagnosis, you neither cite any author. I think they must be named.”
Authors’ comment: We thank the reviewer for his/her helpful comments.
We have inserted the bibliography as reported.
Reviewer’s comment: “You Indicate that: “to isolate the DNA than would allow us to personally identify the subject to whom the skeletal remains had belonged in life, two teeth (34, 37) and a 1cm piece of the femoral shaft of the skeleton were removed” (line 130). Nevertheless, on line 136, you indicate than only “tooth 37 and the vaginal swab were subjected to DNA extraction…”. The other skeletal pieces (tooth 34 and piece of femur) were not analyzed? Nevertheless, on the Discussion you indicate than the both dental pieces were analyzed. It is so confusing. I think it must be clarified.”
Authors’ comment: We thank the reviewer for the comment and the possibility of clarification. Regarding the question asked, we could not use the femur because the magistrate who treated the case asked to keep it for further investigation and tooth 34, particularly for a possible reconstruction of the dental formula. However, the genetic profile extracted twice from tooth 37 was considered sufficient by the entire laboratory team to obtain ten markers as international recommendations on DNA extraction methods needed to identify a subject.
Subsequently, a vaginal swab analysis was performed according to standard decontamination protocols.
We specified this limit in the discussions because the text did not explain it well as follows: “In particular, one of the limitations of our study was the limitations imposed by the magistrate who took care of the case because he expressly requested to protect the most significant number of samples for any subsequent forensic investigation.”
Reviewer’s comment: “About the skeletal remains genetic analysis: On material and methods you indicate that the methodology used to extract DNA from the sample 37 and the vaginal swab was the same. It is very surprising to me, so to the DNA extraction from teeth or bones, usually is necessary to apply specific conditions and also specific cares to minimize contamination. You do not specify what kind of authenticity criteria have used to ensure the absence of exogenous contaminant DNA in the case of the critical skeletal samples. This kind of samples usually are analyzed in specific isolated laboratories, where are no analyzed other kind of well-preserved samples. The sample 37 and the vaginal swab were analyzed simultaneously in the same laboratory? According to the material and methods description it is not clear, and it could be interpreted as a possible source of contamination for the sample 37 (furthermore if the results were no replicated with the sample 34 or the femur). In my opinion, there are too many things to clarify in material and methods of the genetic analysis and the authenticity criteria considered to skeletal remains analysis.”
Authors’ comment: We thank the reviewer for the doubt he raised on the issue; as already mentioned, DNA extraction was performed on tooth 37 by two replicates of PCR. About the extraction from the vaginal swab, it was made in later times thanks to investigations carried out by the Judicial Police and in a separate section of the laboratory after decontaminating the environments as required by the guidelines in Italy. However, in the materials and methods section, this was taken for granted, which is why we have integrated as follows: “In particular, in the preliminary stages of the investigation, the DNA was extracted and examined starting from tooth 37; after the investigations conducted by the judicial police, it was possible to access the vaginal tampon. DNA extraction from the vaginal swab was performed individually and subsequently by a female operator with all the PPE provided and after decontamination of worktops, pipette tips, and pipes, respectively, with exposure to UV rays, sodium hypochlorite, and ethanol.”
Reviewer’s comment:“If there were analyzed more skeletal samples, how many extractions were peformed? And, how many PCRs from each extract?”
Authors’ comment: The DNA was extracted from a single dental element (No. 37) by a double PCR. Following the investigations carried out by the judicial police, it was also possible to access the vaginal tampon. The extraction of DNA from the vaginal swab was carried out in another area of the genetics laboratory before the availability of all preventive measures of risk of contamination in compliance with the guidelines in Italy.
Reviewer’s comment: “(Line 167) about the first skeletal remains finding, I would like it there were performed some dating method to ensure that the remains are contemporary”
Authors’ comment: We want to thank the reviewer for highlighting this aspect, which could not be very clear to the reader. We have specified what was reported as follows:
Reviewer’s comment:“About the results obtained, on Table 2: - I think that it would be explained how were stablished the consensus DNA profile in the case of the skeleton. If there were obtained by different PCRs from different samples (34, 37 and femur), How the consensus was stablished? -What was the criteria to define homozygotes? You consider that there was an allelic dropout phenomenon on: D7S820 and TPOX. Why the cases of: D5S818, and VWA are not also allelic dropout phenome? What was the criteria to stablish these homozygotes?”
Authors’ comment: We are grateful to the reviewer for displaying a confusing element for readers that we have modified as follows: “Only peaks with RFU> 50 were considered, and the drop-out in D7S820 and TPOX markers is a hypothesis supported by the fact that TPOX showed a peak at allele 9 with RFU < 50 in both replicates of DNA PCR extracted from tooth 37. Since no samples of non-degraded tissue were available, it was not possible to confirm abandonment or homozygous in other samples.”
Round 2
Reviewer 1 Report
The manuscript is revised according to the comment.Author Response
Thank you very much
Reviewer 2 Report
I still do not see the methodology related to the replication of results through the analysis of several skeletal samples. Either it has not been done or is not indicated in the text. If it has not been done, it is a mistake on the base of the study.
On line 170 the cite [39] is wrong.
Author Response
Dear Reviewer n'2, Thanks again for your attention to our manuscript to improve quality. Below we report all the news that we have detailed for the execution of genetic investigations and that can be used in the future for possible replication of the results. We have also corrected the inaccuracy of reference number 39. We apologize. We hope that our manuscript will now be appreciated.
"
2.3 Investigations of Forensic Genetics
To isolate the DNA that would allow us to personally identify the subject to whom the skeletal remains had belonged in life, two teeth (34, 37) and a 1 cm piece of the femoral shaft of the skeleton were removed.
The vaginal swab taken during a previous judicial investigation into a case of sexual violence and filed correctly in the refrigerators of the Scientific Police was later used for comparative purposes.
Tooth 37 and the vaginal swab were subjected to DNA extraction at the Forensic Genetics laboratory of the University of Bari using the commercial kit NucleoSpin®DNA Tissue from the company MACHEREY-NAGEL following the specific protocols indicated by the manufacturer [19].
In particular, in the preliminary stages of the investigation, the DNA was extracted and examined starting from tooth 37; after the investigations conducted by the judicial police, it was possible to access the vaginal tampon. DNA extraction from the vaginal swab was performed individually and subsequently by a female operator with all the PPE provided and after decontamination of worktops, pipette tips, and pipes, respectively, with exposure to UV rays, sodium hypochlorite, and ethanol [37].
Amplification of loci D8S1179, D21S11, D7S820, CSF1PO, D3S1358, TH01, D13S317, D16S539, D2S1338, D19S433, vWa, TPOX, D18S51, D5S818, FGA, and Amelogenin were both performed in a multiplex repeat amplification reaction twice, using the reagents of the Applied Biosystems “AmpFLSTR ™ Identifiler ™ Plus PCR Amplification Kit” following the manufacturer's instructions [20].
After amplification, an aliquot of each sample equal to 1µl of the amplified product, after denaturation at 95 ° C for 3 minutes in the presence of 24.5µl of formamide and 0.5µl of an internal standard, was subjected to capillary electrophoresis in an ABI PRISM310 Genetic Analyzer automatic sequencer, Applied Biosystem [21].
The identification of the genetic characteristics of the samples in question, relating to the DNA polymorphisms sought on the autosomal chromosomes, was made with the help of an allelic ladder containing the main Caucasian variants.
The statistical surveys used were the following:
1) calculation of the Random Match Probability (RMP) corresponds to the estimate of the frequency with which a profile occurs in the population.
2) Likelihood ratio (LR) to compare the probability of observing a particular event under two alternative hypotheses, namely Hp: the DNA belongs to that subject, Hd: the DNA does not belong to that subject [2,22].
All procedures have been performed in compliance with SIGU guidelines to prevent possible contamination of samples [39]."